# AnyLayout: Versatile Advertising Poster Layout Generation with MLLMs

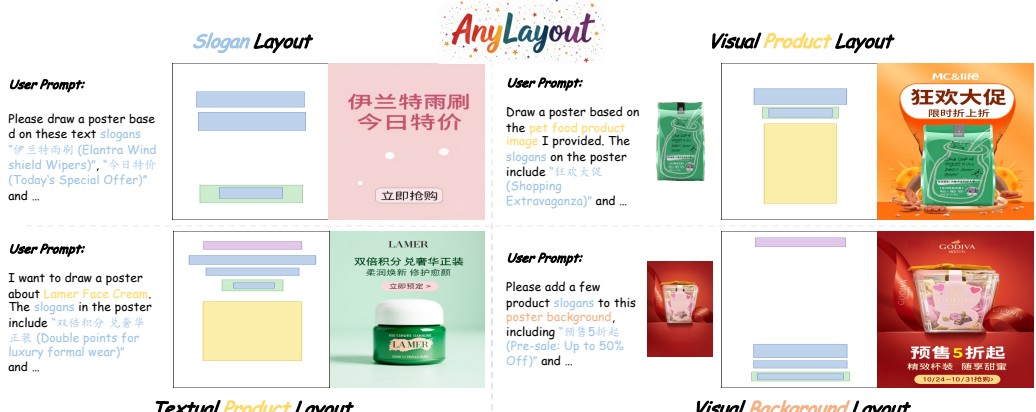

Figure 1: Illustration of our novel, versatile layout generation task: AnyLayout (with four subtasks).

## ABSTRACT

Layout design is a fundamental aspect of visual communication, widely used in advertising, publishing, and digital media. Recent datasets and methods, including content-agnostic and content-aware approaches, have advanced automatic layout generation, and large language models (LLMs) and multi-modal LLMs (MLLMs) have further improved performance. However, most existing methods focus on predicting bounding boxes for limited design elements on fixed backgrounds, which restricts their capability to tackle diverse instruction-driven tasks in real-world applications. To address these limitations, we introduce **AnyLayout-120K**, a large-scale instruction-driven dataset for multimodal layout generation. It offers: (1) *Task Diversity*—comprising four instruction-driven sub-tasks that encompass multimodal design elements such as multi-lingual text, visual/textual product, logos and background underlays; (2) *Rich Annotations*—including user instructions, multimodal inputs and spatial annotations; (3) *Downstream Compatibility*—where, in addition to the layout of individual elements, we propose composite layouts that capture the overall design, integrating both details and semantics. These composite layouts can be seamlessly incorporated into text-to-image (T2I) models for end-to-end generation. Alongside this dataset, we develop 7 geometry-aware evaluation metrics that assess spatial precision and adherence to design principles, ensuring a more comprehensive evaluation. Furthermore, utilizing this dataset, we establish a strong baseline based on MLLMs, achieving state-of-the-art performance. The dataset, metrics, and baseline will be released to support future research in instruction-driven layout design.

## 1 INTRODUCTION

Layout design serves as a fundamental element of visual communication, with essential applications across various domains, including advertising, publishing, digital media, and information design (Yang et al., 2016). Advancements in this field have been marked by the development of notable datasets such as CGL and PKU (Zhou et al., 2022; Hsu et al., 2023), alongside a variety of methods

that encompass both content-agnostic (Li et al., 2019; Chakraborty et al., 2022; Melendez & Havas, 2025) and content-aware (Zheng et al., 2019; Zhang et al., 2025b; Pu et al., 2025) approaches. Recently, there has been an increasing trend toward utilizing LLMs or MLLMs (Hurst et al., 2024; Chu et al., 2024; Chen et al., 2025b; Lu et al., 2024) for automatic layout design (Hsu & Peng, 2025; Cheng et al., 2024; Qu et al., 2025), leading to remarkable improvements in performance.

Despite these advancements, existing methods primarily focus on one single task: predicting bounding boxes for a limited set of design elements on a given background image. Consequently, they struggle to handle various user design requirements in real-world applications. For instance, user instructions such as *"Given an image of a face cream product, design a poster featuring the product on a pink background with a 'best-seller' slogan"* still present great challenges. Such open-ended user instructions that specify multiple design elements – including product images, slogans, and background specifications – highlight the necessity for more versatile layout generation.

In this paper, we extend the field of layout design to more flexible and practical settings, and introduce **AnyLayout-120K**, a large-scale instruction-driven layout dataset. This dataset advances the field through three key innovations as shown in Tab. 1: (1) **Task Diversity**. It encompasses four distinct sub-tasks represented by user instructions, which incorporates multiple interleaved multi-modal design elements (e.g., text, logos in two languages, and underlays). (2) **Rich Annotations**. In addition to the instructions, multimodal design elements, and traditional spatial annotations (e.g., bounding boxes for individual elements), it also provides structured natural language descriptions of *composite* layouts — capturing both element-level details and overall design semantics. (3) **Downstream Compatibility**. The structured layout descriptions can be seamlessly integrated with T2I models, facilitating end-to-end content generation. Based on this dataset, we develop an enhanced evaluation system comprising 7 *geometry-aware evaluation metrics* that move past conventional IoU-style scoring. These metrics quantify both spatial precision and adherence to design principles (e.g., utilization, non-occlusion), thus ensuring a more comprehensive evaluation.

Furthermore, engaging with AnyLayout-120K presents challenges that require deep multi-modal understanding: the model must effectively align visual and textual semantics, adhere geometric constraints, and produce coherent and appealing layouts under diverse conditions. To address these challenges, we propose a unified MLLM-based layout model that generates layouts in natural language formats, simultaneously optimizing composite and individual elements, ensuring geometric plausibility and visual harmony.

In summary, our main contributions are as follows.

(i) We introduce **AnyLayout-120K**, a large-scale instruction-driven layout dataset that features four sub-tasks composed of user instructions and interleaved multimodal design elements. Alongside this dataset, we propose 7 geometry-aware metrics to ensure a more comprehensive assessment.

(ii) In addition to providing placements for individual design elements, we propose composite layouts that describe the overall design layout. These composite layouts can be seamlessly integrated into T2I models, facilitating the effective rendering of the generated designs.

(iii) We establish a strong baseline that achieves state-of-the-art performance across tasks, providing a foundation for future research in instruction-driven layout design.

## 2 RELATED WORK

**Content-Agnostic Layout Generation.** Early work abstracts layouts into elements with categorical labels and geometric parameters, focusing on structural and spatial alignment while ignoring semantic context. Representative models include GAN-based LayoutGAN (Li et al., 2019), VAE-based LayoutVAE (Jyothi et al., 2019), and Transformer–VAE hybrids such as VTN (Arroyo et al., 2021) and bidirectional masked BLT (Kong et al., 2022), which balance global alignment with diversity. Beyond diffusion models, Flow Matching approaches like LayoutFlow (Guerreiro et al., 2024) and discrete diffusion with external correction as in Layout-Corrector (Iwai et al., 2024) improve convergence stability and geometric controllability. While these paradigms provide strong structural priors, they lack multimodal semantic adaptation needed in real-world advertising and poster design.

**Content-Aware Visual–Textual Layout.** Content-aware methods tailor layouts to specific inputs such as products, slogans, and backgrounds. CGL-GAN (Zhou et al., 2022) generates design layouts from image composition and introduces metrics aligned with aesthetic intuition. PosterLay-

out (Hsu et al., 2023) models non-empty canvases and design order, establishing benchmarks and evaluation criteria. AutoPoster (Lin et al., 2023) explores human–AI co-creation workflows for advertising posters. LayoutFormer++ (Jiang et al., 2023) unifies multiple conditional tasks via constraint serialization, while DETR-based LayoutDETR (Yu et al., 2024) demonstrates the strength of detection-style representations for multimodal conditional layouts. Although these works have enriched datasets and metrics, most assume fixed or partially pre-filled canvases, limiting generalization to common scenarios like "pure slogan", "single product", or "background-only" inputs.

**Layout Generation with MLLMs.** Recent work leverages LLMs and MLLMs to convert layouts into structured, executable formats (e.g., HTML/JSON), enhancing interpretability and consistency. LayoutNUWA (Tang et al., 2023) pioneered "code-based" layouts with improved semantic alignment via instruction tuning. PosterLlama (Seol et al., 2024) and PosterLLaVA (Yang et al., 2024) adapt layout generation into LLM/MLLM pipelines, enabling natural language constraints, editable SVGs, and multimodal interaction. VASCAR (Zhang et al., 2024) iteratively refines layouts through visual self-correction in LVLMs. Strong multimodal base models such as Qwen2.5-VL (Bai et al., 2025)—with fine-grained localization and document/graph parsing capabilities—form the infrastructure for end-to-end instruction-to-layout training and evaluation. Aligned with our work, these approaches advocate a *unified task interface* plus *executable structural outputs*, paving the way for cross-task alignment and downstream renderability.

## 3 ANYLAYOUT DATASET

We extend layout generation from fixed-canvas settings to a *versatile, instruction-driven* formulation that supports multimodal inputs (*slogan-only*, *textual product*, *visual product*, *visual background*). Specifically, we contribute: (i) a unified task interface covering the four sub-tasks (Fig. 1), (ii) AnyLayout-120K, a large-scale instruction-driven dataset and (iii) geometry-aware, task-conditioned metrics that enhance the prior assessments used in CGL/PKU. This section is organized as follows: Sec. 3.1 introduces the four sub-tasks; Sec. 3.2 presents product-centered metrics and Sec. 3.3 outlines a four-stage data construction pipeline.

### 3.1 PROPOSED TASK

Given an optional pair of images (product/background) and the instructions input, the task is required to output not only the placement of individual design elements, but also the composite layout to support downstream T2I model to generate poster. We therefore cast layout generation as an instruction-driven design problem, closer to real practice than fixed-canvas formulations. As illustrated in Fig. 1, we define four sub-tasks: (1) *Slogan Layout*, (2) *Textual Product Layout*, (3) *Visual Product Layout*, and (4) *Visual Background Layout*.

*Slogan Layout.* This task aims to generate optimal poster layouts from textual slogans, particularly for cases containing only text, such as exhibition themes or cultural promotion posters. The task requires the text's position, scale, and arrangement solely based on the given slogan.

*Textual Product Layout.* This task addresses practical poster-generation needs where layouts are derived solely from textual descriptions of a product, complemented by slogans or other elements. For example, it may involve creating a face cream-selling poster featuring a green-scene background and a specific slogan. Effective design in such cases requires accurately interpreting the concepts in the text and arranging the elements with precise spatial organization.

*Visual Product Layout.* Building on the product-description task, a more constrained and practical scenario incorporates the product image alongside the text slogan as input. The task requires a poster layout that seamlessly integrates visual and textual elements, aligning layout geometry with semantic content. This necessitate reasoning about relative positioning, scale, and visual hierarchy under the constraints of real image inputs.

*Visual Background Layout.* A common application scenario involves incorporating a text slogan layout frame into a poster image that already contains a primary product and background. In our proposed task, the input consists solely of the slogan, without specifying the dimensions of the text layout or providing category and size information for any additional layout elements to be predicted.

| Dataset | Instr. Driven | Input Design Elements | Output | Slogan Lang. | Aspect Ratio |
|---|---|---|---|---|---|
| CGL | ✗ | Back.Img(Op) & Cate/Size | Layout | CN | (2:3) |
| AP | ✗ | Back.Img(Op) & Cate/Size | Layout | CN | (2:3) |
| PKU | ✗ | Back.Img(Op) & Cate/Size | Layout | CN | (2:3) |
| **AnyLayout-120K** | ✓ | Back/Prod.Img(Op) & Instruction | Layout & Composite | CN, EN | (2:3),(1:1) |

Table 1: Comparison of AnyLayout-120K with existing poster layout datasets. *Instr. Driven* denotes support for explicit, user instruction–driven generation. *Input Design Elements* describes the optional or required inputs: *Back.Img* means a poster background with product already placed, *Op* = optional, *Cate/Size* = assigned category or bounding-box size, *Back/Prod.Img(Op)* = poster background or product image provided optionally, plus instructions, categories, and slogans. *Output* lists what the model generates: conventional layouts or extended composite layouts combining semantic and geometric information. *Slogan Lang.* shows supported languages for slogans. *Aspect Ratio* lists available layout aspect ratios.

## 3.2 PRODUCT-CENTERED METRICS

We introduce new evaluation metrics from a product-centered perspective, as existing content-layout benchmarks do not involve product placement and interaction. Drawing inspiration from typographic and layout studies (Ma et al., 2024; Rebelo et al., 2024) as well as established graphic design principles (Ngo et al., 2000; Harrington et al., 2004), we propose task-conditioned, geometry-aware metrics. These metrics aim to provide a more comprehensive assessment of layout designs, specifically focusing on the effective integration of products within the overall composition.

Each metric is a *geometry functional* that operates solely on predicted and ground-truth box coordinates. The metrics include: Centrality Score ($CS$), Size Ratio Norm ($SR_{\text{Norm}}$), Overlap Score ($OS$), Vertical Position Score ($VPS$), Pair Distance Score ($PDS$), Dispersion Consistency Score ($DCS$), and Size Consistency Score ($SCS$). Together, these metrics effectively capture both aesthetic alignment (e.g., balance and hierarchy) and functional positioning (e.g., occlusion avoidance and scale consistency).

The $CS$ measures how close the product's center is to the poster's center, with higher scores awarded for more central placements:

$$CS = 1 - \frac{d}{d_{max}}$$
$$d_{max} = \sqrt{(W/2)^2 + (H/2)^2} \tag{1}$$
$$d = \sqrt{(c_x - W/2)^2 + (c_y - H/2)^2}$$

where $(c_x, c_y)$ is the product-box center. If $d=0$, then $CS=1$; if $d=d_{max}$, then $CS=0$.

The $SR_{\text{Norm}}$ encourages a product area within a desirable range; overly small boxes are penalized linearly, while overly large boxes are saturated at the upper bound:

$$SR_{\text{Norm}} = \frac{min(\frac{A_{\text{prod}}}{A_{\text{img}}}, S_{max})}{S_{max}}$$
$$A_{\text{prod}} = (x_2 - x_1)(y_2 - y_1) \tag{2}$$
$$A_{\text{img}} = W \times H$$

here $(x_1, y_1)$ and $(x_2, y_2)$ are the top-left and bottom-right corners of the product box, and $S_{\text{max}}$ is the upper-area threshold.

The $OS$ penalizes occlusion between the product and other elements (text, underlay, etc.):

$$OS = 1 - \frac{\sum_i \text{Area of Intersection}(prod, other_i)}{A_{prod}} \tag{3}$$

The $VPS$ encourages the product to sit near a task-specific vertical target (e.g., lower-center emphasis); $t \in [0, 1]$ controls the preferred normalized vertical position:

$$VPS = 1 - \frac{\frac{c_y}{H} - t}{max(t, 1 - t)} \tag{4}$$

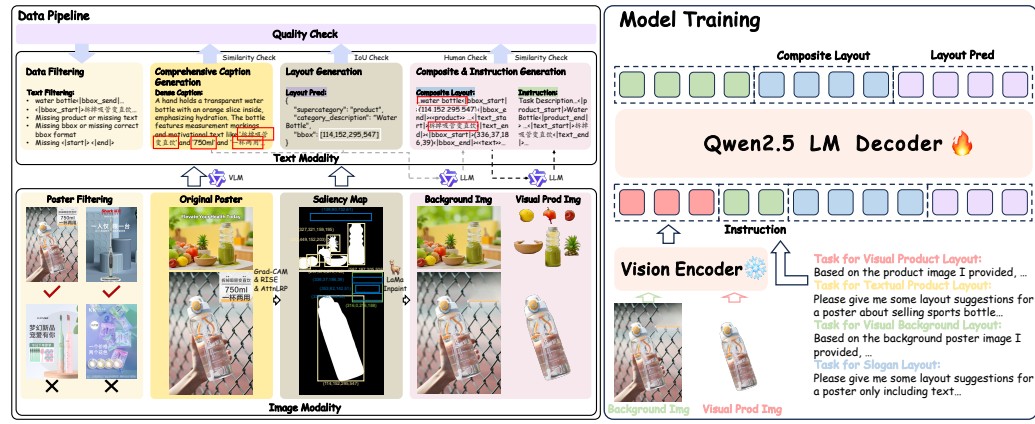

Figure 2: Our data processing pipeline for image and text modality inputs, along with the model training framework.

For multi-product cases, promotes reasonable spacing by averaging pairwise center distances–$PDS$ (normalized by the image diagonal):

$$PDS = \frac{1}{N} \sum_{i<j} \frac{d_{ij}}{\sqrt{W^2 + H^2}} \tag{5}$$

where $d_{ij}$ is the Euclidean distance between centers of boxes $i$ and $j$, and the sum runs over all unordered pairs (the $1/N$ factor denotes averaging).

The $DCS$ measures the uniformity of inter-product spacing via the coefficient of variation:

$$DCS = 1 - \mathcal{CV}(\{\frac{d_{ij}}{\sqrt{W^2 + H^2}}\}), \tag{6}$$

where $\mathcal{CV} = \sigma/\mu$ is the coefficient of variation. More uniform distances (smaller $\mathcal{CV}$) yield higher scores.

The $SCS$ encourages comparable product scales by penalizing variation in area ratios:

$$SCS = 1 - \mathcal{CV}(\{\frac{A_i}{WH}\}_{i=1}^N) \tag{7}$$

### 3.3 DATA PIPELINE

To support the four instruction-driven tasks in Sec. 3.1, we build a systematic, mutually validating dataset pipeline (Fig. 2) with **five** stages: **(1) Data Filtering**, **(2) Comprehensive Caption Generation**, **(3) Layout Generation**, **(4) Composite & Instruction Generation**, and **(5) Quality Check**. This structure mirrors our product-centered metrics in Sec. 3.2, ensuring that data construction, validation, and evaluation use consistent geometry-aware criteria.

To this end, we address the scarcity of *instruction-driven* resources for controllable poster layout by converting four heterogeneous datasets—PKU (Hsu et al., 2023), AutoPoster (AP) (Lin et al., 2023), CGL (Li et al., 2023), and CreatiDesign (CD) (Zhang et al., 2025a)—into a unified dataset tailored for multimodal *instruction-driven* fine-tuning. Beyond raw coordinates, each sample is augmented with a complete *instruction–driven* pair: natural-language task description, multimodal context, and a machine-verifiable structured answer.

**Data Filtering.** To remove annotation noise in multi-product posters from PKU/AP/CGL, we retained 60K single-product samples via Data Filtering. For the multi-product, 60K samples were randomly selected from CD. These two subsets were then combined to form the original 120K-poster layout dataset. Furthermore, Data Filtering employs a simultaneous filtration process on the data subsequent to Quality Check, ensuring the establishment of a high-quality dataset.

**Comprehensive Caption Generation & Layout Generation.** Following GoT (Fang et al., 2025), we pair boxes with descriptive captions encoding spatial and semantic layout information. We implement captioning with Qwen2.5-VL (Bai et al., 2025) using a prompt that covers: (1) fine-grained

product description, (2) background composition/style, (3) typography/decorations, and (4) slogan content/placement. To correct occasional slogan errors, we align generated captions to ground-truth slogans via semantic similarity matching, repair mismatches, and filter low-alignment cases (Fig. 2).

For Layout Generation, text/background boxes are taken from original labels. For visual-product localization, we compute *model-agnostic* saliency and convert it to boxes via morphology and connected components: apply Grad-CAM (Selvaraju et al., 2017) on a product-recognition encoder and cross-validate with RISE (Petsiuk et al., 2018) to reduce model-specific bias; threshold the consensus map, extract the maximal connected component, and fit a tight box. Low-confidence or multi-peak cases fall back to attention-rollout via AttnLRP (Chefer et al., 2021) to preserve recall. We binarize with Otsu+area priors, generate candidates, and select the top-confidence box with non-maximum suppression. This yields accurate product localization (Fig. 2), and—together with refined captions—forms a compact, executable multimodal annotation.

**Composite & Instruction Generation.** After captioning and layout generation, we refine captions with Qwen3 (Yang et al., 2025) to inject explicit layout references then gain the Layout-Aware Composite. The module aligns caption spans to regions by semantic similarity and replaces matched phrases with structured placeholders $< |box\_start| > \ldots < |box\_end| > \langle\langle\texttt{product/logo/text/underlay/embellishment}\rangle\rangle$ (Fig. 2), enabling precise text–geometry linkage.

For instruction generation, we collected high-quality user input based on actual design requirements and had Qwen3 refine it based on the composite. This refinement retained the necessary visual product and slogan layout elements, as well as the poster aspect ratio, within the instructions. The layout elements were then matched with the special symbols $< |product\_start| > \ldots < |product\_end| >$ and $< |text\_start| > \ldots < |text\_end| >$, respectively, to complete the different instruction set for four task scenarios (see the Appendix for more instruction set examples).

Both Composite generation and Instructions generation will undergo similarity check for the visual product and slogan first. Then we quantify quality via human check under a stratified protocol. We sample $n=400$ items from AnyLayout-120K, stratified by the four tasks ($4 \times 100$) and by single/multi-product (50/50). The reviewer judges (i) *box correctness* (product/text/underlay acceptable if IoU to intended region $\geq 0.7$ or justified tightness) and (ii) *caption–box alignment*.

**Quality Check.** To estimate datasets quality including VLM/LLM generation results and visual product layout generation results, we deploy a deterministic validator consisting of *Similarity Check*, *IoU Check* and *Human Check*:

- *Similarity Check*: we apply it on *Comprehensive Caption Generation* and *Composite & Instruction Generation*. It is used to check the VLM/LLM generated slogan and the visual product whether match the ground-truth.

- *IoU Check*: we mainly utilize it on *Layout Generation*. Then, we check the visual product location quality by requiring IoU ($>0.6$) agreement between Grad-CAM and RISE otherwise fall back to attention-rollout AttnLRP.

- *Human Check*: we let human reviewer to check a gold sample of Composite and Instruction generated by LLM whether the layout element match the correct slogan or visual product.

Across Quality Check, the *Similarity Check* and *IoU check* accepts 92.5% ($\pm 2.1\%$) of samples; among these, the *Human Check* confirms 95.1% ($\pm 1.9\%$) box correctness and 96.4% ($\pm 2.0\%$) caption–box alignment. Combining (i) *Similarity & IoU* pass and (ii) *Human Check* pass yields an end-to-end clean-label estimate of 93.7% ($\pm 2.9\%$).

# 4 METHOD

## 4.1 UNIFIED LAYOUT MODEL

Recent MLLM studies report emergent layout understanding and the ability to interpret spatial coordinates for coherent layout generation (Seol et al., 2024; Zhang et al., 2024; Chen et al., 2025a; Tang et al., 2023). To validate our task design and the utility of AnyLayout-120K, we build an SFT baseline on Qwen2.5VL-7B.

**Single-task SFT.** We first fine-tune four *single-task* models. Each output is a JSON-like sequence containing *category*, *short description*, and *bbox* for every element. Training minimizes the next-token cross-entropy:

$$\mathcal{L}_{CE} = -\sum_{i=1}^{N} y_i \log(p_i) \tag{8}$$

computed over the serialized instruction–response.

**Unified multi-task SFT.** We then train a *single* model across all four tasks using a balanced 1:1:1:1 sampler with randomized batch shuffling. This exposes the model to heterogeneous input regimes (*slogan-only*, *textual/visual product*, *visual background*) and encourages transfer of geometry–semantics priors across tasks. In practice, joint training yields consistently better average performance than isolated training, indicating useful cross-task synergies.

## 4.2 Composite Layout Prediction

Layout prediction requires reasoning over *what* (semantics) and *where/how* (geometry, scale, hierarchy) jointly. Rather than designing task-specific heads, we serialize an *executable composite layout* that captures element types, semantic attributes, and inter-element geometric relations in a single sequence. This reframes the problem from independent coordinate regression to autoregressive *joint geometry–semantics* reasoning. For example, two elements—a visual product "*face cream*" and a slogan "*Double points for luxury formal wear*"—are emitted as a compact description with their bboxes embedded at the points where they are referenced.

Compared with composite-oriented frameworks such as PosterLlama (Seol et al., 2024), Poster-LLaVA (Yang et al., 2024), and LayoutNUWA (Tang et al., 2023), our formulation differs in two geometric aspects: (i) we *merge* textual semantics and coordinates into one executable sequence that encodes cross-element constraints, enabling stepwise reasoning in a single state space; (ii) the same composite space is *instruction-compatible* across all input modalities (*slogan-only*, *textual/visual product*, *visual background*), avoiding fixed-canvas or placeholder-only assumptions and eliminating task-specific decoders.

All four subtasks are projected into this common composite space, allowing the unified model to learn transferable design priors. The representation serves as a *binding contract* to downstream renderers: predicted sequences can be directly executed to produce layouts. While our baseline uses Qwen2.5-VL-7B with instruction tuning, the formulation is model-agnostic and invites future architectures tailored for composite layout reasoning.

# 5 Experiment

## 5.1 Experiment Settings

**Datasets.** We evaluate on **AnyLayout**, comprising five element types—logo, text, product, underlay, and embellishment. The dataset contains 126,131 annotated *poster–layout* pairs, split into 118,450 for training and 7,681 for testing. Following our data pipeline, AnyLayout aggregates four sources: CGL (train 20,851 / test 1,026), AutoPoster (train 31,495 / test 3,655), PKU (train 6,726), and CreatiDesign (train 59,378 / test 3,000). Training data covers four novel tasks with product/text annotations (including product and background images), while the test set contains annotated samples for each task.

**Evaluation Metrics.** We adopt: (1) PKU (Hsu et al., 2023) and CGL (Li et al., 2023) content-aware layout benchmarks; (2) a new single-/multi-product benchmark. PKU metrics include $ali$, $und_l$, $und_s$, $ove$, and $val$; CGL metrics comprise $R_{ove}$, $R_{und}$, $R_{ali}$, and $R_{occ}$. $R_{occ}$ / $val$ measure unused or invalid layout space, $R_{ali}$ / $ali$ assess element alignment, $ove$ / $R_{ove}$ (via IoU) quantify non-decorative element overlap, and $R_{und}$ / $und_l$ / $und_s$ evaluate the enhancement from decorative to non-decorative elements.

**Baseline.** Our goal is to validate the *AnyLayout* task, dataset, and unified benchmarks. We benchmark mainstream multimodal layout predictors—PosterLlama (Seol et al., 2024), Poster-LLaVA (Yang et al., 2024)—in zero-shot mode (due to mismatched I/O formats) and compare with Qwen2.5VL-7B (Bai et al., 2025) zero-shot outputs, serving as a baseline for our SFT and *SFT w/ C* approaches.

| Tasks | | PKU Metrics | | | | | CGL Metrics | | | |
|---|---|---|---|---|---|---|---|---|---|---|
| Methods | Tasks | $ali\downarrow$ | $und_l\uparrow$ | $und_s\uparrow$ | $ove\downarrow$ | $val\uparrow$ | $R_{ove}\downarrow$ | $R_{und}\uparrow$ | $R_{ali}\downarrow$ | $R_{occ}\uparrow$ |
| Zero-shot | Slogan Layout | 0.0236 | 0.8451 | 0.3485 | 0.2778 | **1.0** | 0.4637 | 0.8523 | 0.0264 | 0.7139 |
| SFT w/ C | Slogan Layout | 0.0241 | 0.9863 | 0.9775 | 0.0038 | 0.9998 | 0.0093 | 0.9896 | 0.0148 | **0.9644** |
| SFT w/ C | Mix | 0.0208 | **0.9951** | **0.9900** | **0.0021** | 0.9998 | 0.0050 | **0.9965** | 0.0098 | **0.9644** |
| SFT | | **0.0160** | 0.9901 | 0.9875 | 0.0023 | 0.9997 | **0.0042** | 0.9922 | **0.0030** | 0.9621 |
| PosterLlama | | 0.0229 | 0.8681 | 0.5550 | 0.2677 | 0.9995 | 0.4526 | 0.8613 | 0.0252 | 0.8250 |
| PosterLLaVA | | 0.0214 | 0.8857 | 0.5679 | 0.2523 | 0.9996 | 0.4482 | 0.8766 | 0.0241 | 0.8357 |
| Zero-shot | Txt Prod Layout | 0.0225 | 0.9334 | 0.9301 | 0.0222 | **1.0** | 0.0486 | 0.9336 | 0.0046 | 0.8848 |
| SFT w/ C | Txt Prod Layout | **0.0168** | 0.9818 | 0.9688 | 0.0123 | 0.9998 | 0.0277 | 0.9813 | 0.0050 | **0.9869** |
| SFT w/ C | Mix | 0.0229 | **0.9829** | **0.9787** | **0.0021** | 0.9999 | **0.0043** | 0.9844 | **0.0039** | 0.9836 |
| SFT | | 0.0195 | 0.9803 | 0.9767 | 0.0032 | 0.9997 | 0.0056 | **0.9928** | 0.0040 | 0.9712 |
| PosterLlama | | 0.0218 | 0.9002 | 0.8915 | 0.0206 | 0.9996 | 0.0469 | 0.8987 | 0.0063 | 0.8991 |
| PosterLLaVA | | 0.0210 | 0.9018 | 0.8874 | 0.0194 | 0.9999 | 0.0435 | 0.9011 | 0.0051 | 0.9008 |
| Zero-shot | Vis Prod Layout | 0.1000 | 0.7860 | 0.6407 | 0.2759 | **1.0** | 0.3797 | 0.7912 | 0.0325 | 0.8410 |
| SFT w/ C | Vis Prod Layout | 0.0243 | 0.9777 | 0.9701 | 0.0104 | 0.9999 | 0.0149 | 0.9777 | 0.0070 | 0.9817 |
| SFT w/ C | Mix | 0.0224 | **0.9916** | **0.9886** | 0.0024 | 0.9999 | **0.0052** | **0.9917** | 0.0062 | **0.9823** |
| SFT | | **0.0193** | 0.9877 | 0.9852 | 0.0032 | 0.9997 | 0.0054 | 0.9915 | 0.0069 | 0.9689 |
| PosterLlama | | 0.0872 | 0.8016 | 0.6527 | 0.2431 | 0.9995 | 0.3369 | 0.8054 | 0.0258 | 0.8722 |
| PosterLLaVA | | 0.0695 | 0.8136 | 0.6473 | 0.2289 | 0.9996 | 0.3077 | 0.8152 | 0.0258 | 0.8814 |
| Zero-shot | Vis Bg Layout | 0.0463 | 0.6442 | 0.4758 | 0.2262 | **1.0** | 0.3657 | 0.6462 | 0.0172 | 0.8686 |
| SFT w/ C | Vis Bg Layout | **0.0126** | 0.9689 | 0.9612 | 0.0140 | 0.9996 | 0.0206 | 0.9739 | 0.0058 | **0.9802** |
| SFT w/ C | Mix | 0.0134 | **0.9908** | **0.9848** | 0.0039 | 0.9997 | **0.0069** | **0.9905** | 0.0038 | **0.9802** |
| SFT | | 0.0131 | 0.9871 | 0.9794 | 0.0060 | 0.9991 | 0.0091 | 0.9890 | 0.0039 | 0.9745 |
| PosterLlama | | 0.0461 | 0.6970 | 0.5489 | 0.2127 | 0.9995 | 0.3599 | 0.6976 | 0.0166 | 0.8710 |
| PosterLLaVA | | 0.0432 | 0.7315 | 0.5264 | 0.2038 | 0.9995 | 0.3571 | 0.7296 | 0.0170 | 0.8826 |

Table 2: Experiment results on AnyLayout-test (w/o product). *Txt*, *Prod*, *Vis*, *Bg* denote *Textual*, *Product*, *Visual*, and *Background*. Best performance per column is in **bold**.

| Tasks | | Single Product Metrics $\uparrow$ | | | | | | Multi Product Metrics $\uparrow$ | | | | | |
|---|---|---|---|---|---|---|---|---|---|---|---|---|---|
| Methods | Tasks | $CS$ | $SR_{\text{Norm}}$ | $OS$ | $VPS$ | $MeanIoU$ | $CPS$ | $CPS$ | $MeanIoU$ | $PDS$ | $DCS$ | $SCS$ | $CPS_m$ |
| Zero-shot | Vis Prod Layout | 0.7488 | **0.9987** | 0.6482 | 0.7830 | 0.4233 | 0.7946 | 0.6182 | 0.2219 | 0.1957 | **0.7821** | 0.3894 | 0.4964 |
| SFT w/ C | Vis Prod Layout | 0.7847 | 0.9605 | 0.9357 | 0.8876 | **0.7512** | 0.8921 | 0.6819 | **0.4332** | **0.2479** | 0.7040 | 0.3820 | **0.5040** |
| SFT w/ C | Mix | **0.7850** | 0.9656 | **0.9424** | 0.8880 | 0.7061 | **0.8952** | **0.6876** | 0.3923 | 0.2375 | 0.7007 | 0.3760 | 0.5005 |
| SFT | | 0.7835 | 0.9644 | 0.9378 | **0.8947** | 0.7274 | 0.8950 | 0.6502 | 0.3030 | 0.2258 | 0.7156 | **0.4087** | 0.5001 |
| PosterLlama | | 0.7415 | 0.9940 | 0.6702 | 0.7917 | 0.4259 | 0.7993 | 0.6257 | 0.2306 | 0.1981 | 0.7124 | 0.3925 | 0.4822 |
| PosterLLaVA | | 0.7541 | 0.9948 | 0.6539 | 0.8002 | 0.4310 | 0.8008 | 0.6284 | 0.2517 | 0.2005 | 0.7294 | 0.3901 | 0.4871 |
| Zero-shot | Txt Prod Layout | 0.8882 | **1.0** | 0.7737 | 0.7940 | 0.4689 | 0.8640 | 0.6074 | 0.1445 | 0.2593 | 0.6054 | 0.4417 | 0.4785 |
| SFT w/ C | Txt Prod Layout | 0.7874 | 0.9543 | 0.9304 | 0.8868 | **0.6264** | 0.8897 | 0.6348 | 0.1985 | 0.2542 | 0.6864 | 0.4395 | 0.5038 |
| SFT w/ C | Mix | 0.7941 | 0.9623 | 0.9293 | **0.8952** | 0.5959 | **0.8953** | **0.6934** | **0.2643** | 0.2465 | **0.6927** | 0.3911 | **0.5059** |
| SFT | | 0.7901 | 0.9639 | **0.9332** | 0.8903 | 0.6060 | 0.8943 | 0.6501 | 0.2074 | 0.2247 | 0.6917 | 0.4312 | 0.4994 |
| PosterLlama | | 0.8820 | 0.9983 | 0.7479 | 0.7825 | 0.4438 | 0.8527 | 0.6095 | 0.1507 | 0.2588 | 0.6106 | 0.4439 | 0.4807 |
| PosterLLaVA | | **0.8952** | 0.9847 | 0.7563 | 0.7801 | 0.4695 | 0.8541 | 0.6194 | 0.1689 | **0.2653** | 0.6055 | **0.4510** | 0.4853 |

Table 3: Comparison of **Single Product** and **Multi Product** metrics for different methods. *Txt*, *Prod*, *Vis* denote *Textual*, *Product*, and *Visual* respectively. Best performance per column is in **bold**.

**Implementation Details.** Based on Qwen-2.5-VL-7B (Bai et al., 2025), we fine-tune our model with the following experiment settings: learning rate of 1.0e-5, global batch size of 16, and image maximum input resolution is set to $1024 \times 1024$ pixels. We utilize LLaMA-Factory (Zheng et al., 2024) as our supervised fine-tuning (SFT) codebase. We perform SFT on our proposed AnyLayout-120K dataset for 3 epochs with 8 NVIDIA H20 GPUs, and the training steps are the same for the model trained with extra reasoning process.

## 5.2 MAIN RESULTS

Tab. 2 reports PKU/CGL results for the four sub-tasks; Tab. 3 gives our single-/multi-product metrics assessing spatial alignment, scale consistency, and inter-object arrangement. Fig. 3 visualizes the full pipeline: AnyLayout predicts category, bounding box, localized text, product description, and a composite layout string; the latter conditions Flux-Kontext (Labs et al., 2025) to render layouts faithful to semantics and spatial constraints.

Relative to zero-shot Qwen2.5VL, all SFT variants achieve substantial gains across PKU ($ali\downarrow$, $und\uparrow$, $ove\downarrow$, $val\uparrow$) and CGL ($R_{ove}\downarrow$, $R_{und}\uparrow$, $R_{ali}\downarrow$, $R_{occ}\uparrow$) for every task. Improvements carry over to product-centric metrics (Tab. 3), where SFT consistently boosts $CS$, $OS$, $VPS$, $MeanIoU$, $CPS$, and their multi-product counterparts.

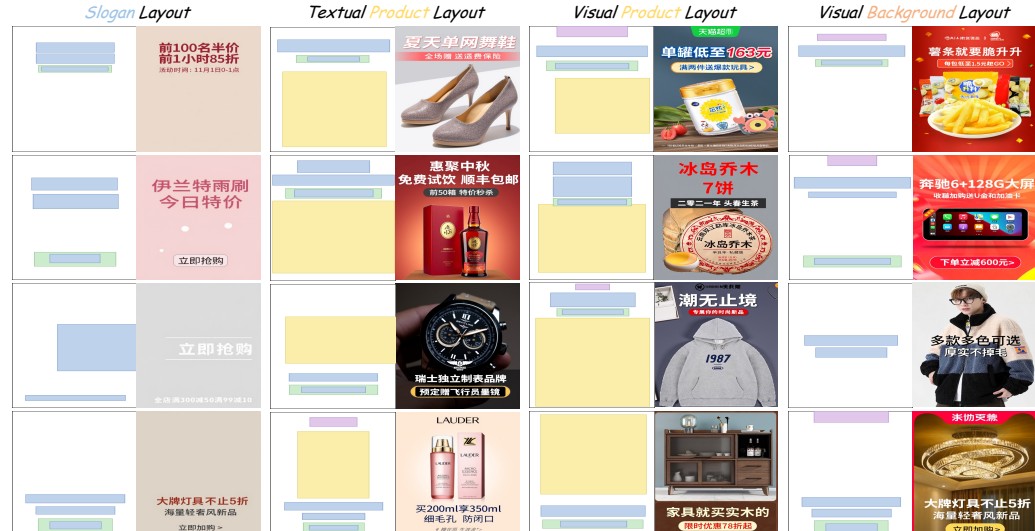

Figure 3: Qualitative results on AnyLayout test set, each sub-task is displayed in a separate column.

Across both PKU and CGL, PosterLlama and PosterLLaVA lag behind any SFT-trained AnyLay-out; PosterLLaVA generally outperforms PosterLlama (e.g., better $und$ and lower $R_{ove}$), with rare reversals. Under our proposed metrics, the same pattern holds—PosterLLaVA slightly leads in $VPS/MeanIoU/CPS$ for *Vis Prod Layout*—yet both are far below SFT models.

System-level ranking is consistent across benchmarks: *SFT w/ C* $\approx$ SFT > zero-shot Qwen2.5VL > PosterLLaVA > PosterLlama, with minor variations on columns like $OS$ or $SCS$. This sta-bility across legacy and proposed metrics indicates the latter capture capability gaps aligned with established criteria while being more sensitive to product placement and compositional fidelity.

### 5.3 ABLATION STUDIES

**Mix Training.** Multi-task training (*SFT w/ C, Mix*) surpasses single-task SFT on most metrics, par-ticularly improving spatial/scale consistency ($VPS$, $MeanIoU$, $CPS$, $CPS_m$) without degrading category or overlap scores, confirming that shared task structure promotes generalization.

**Composite Layout.** Compared to plain SFT, *SFT w/ C* yields consistent PKU gains and dominates on CGL for *Vis Prod Layout* and *Vis Bg Layout*, with improvements in most metrics for *Txt Prod Layout* and *Slogan Layout*. Similar trends appear in Table 3, where composite layouts improve most single-/multi-product metrics. This suggests composite layout strings provide a strong inductive bias for coherent interactions among text, product, and background—advantages retained when rendered with Flux-Kontext (Labs et al., 2025).

## 6 CONCLUSION

In this paper, we propose **AnyLayout-120K**, a comprehensive dataset and benchmark for advertis-ing poster layout generation, which advances the field through four diverse sub-tasks, rich design varieties, and language-conditioned layout prediction. Based on the proposed dataset, we present an MLLM-based model as a strong baseline, which unifies composite and fine-grained spatial reason-ing through natural languages with product or background images as optional visual inputs, enabling coherent and context-aware layout generation. Extensive experiments demonstrate consistent supe-riority over existing methods on different tasks and metrics. In short, AnyLayout establishes a new paradigm for layout modeling by integrating semantic understanding, structural control, and cross-modal generation with one single model, representing a significant step toward scalable, intelligent design automation in complex, real-world scenarios.

## 7 REPRODUCIBILITY STATEMENT

To ensure the full reproducibility of our findings, we have provided comprehensive implementation details throughout the paper. Each four tasks I/O examples to describe our datasets at Sec. 3 and Appendix A.1. Key details of instructions we reformulates them based on four tasks are presented on Sec. 3.3 and Appendix A.2. Moreover, AnyLayout-120K datasets analysis is discussed at Appendix A.3 and baseline architecture of AnyLayout framework is described at Sec. 4. In line with our commitment to open science, AnyLayout-120K dataset and source code will be made publicly available.

## 8 ETHICS STATEMENT

This research adheres to the ICLR Code of Ethics in all aspects of its execution, including data collection, analysis, and dissemination of results. The study does not involve human subjects, animal experiments, or sensitive personal data. All datasets used are publicly available and were collected in compliance with applicable laws and licenses. We have reviewed the datasets to the best of our ability to minimize potential bias, discrimination, or unfairness, and to avoid inclusion of harmful or offensive content.

The methods proposed pose no foreseeable risk to individuals, groups, or the environment, and are intended for academic and socially beneficial purposes. Any potential misuse scenarios have been considered and mitigated through appropriate design choices. No conflicts of interest or sponsorships that could have influenced the results are present. All results are reported honestly, without fabrication, falsification, or inappropriate manipulation, in line with the principles of research integrity.

By including this statement, the authors explicitly acknowledge their obligation to comply with the ICLR Code of Ethics throughout the submission, review, and discussion process.

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
