# A  APPENDIX

## A.1  DATA DESCRIPTION

We define two types of dataset input and output formats for each task: 1) SFT; 2) SFT with Composite.

Here are datasets description examples about four different tasks. We highlight the input aspect-ratio size and different input elements for different tasks, including the product image, background image, product description, and slogan.

**Task for Slogan Layout**

*SFT*

**Input:**
```
Task Description:

Please give me some layout suggestions for drawing a poster
```
**which width and height ratio is 1:1.**  The poster should have the
following text:  **<|text_start|>SWEET & SPICY PORK RIBS<|text_end|>**
**and <|text_start|>Get Saucy!<|text_end|>.**
```
System Prompt:

"NOTE":  "You only need output your answer about layout
generation.  The answer is required by this json format:
"["supercategory":  "text/logo/underlay/embellishment",
"category_description":  "text slogan(text)/none(logo, underlay,
embellishment)", "bbox":  [x, y, w, h],...]."
```
**Output:**
```
["supercategory":  "text", "category_description":  "Get Saucy!",
"bbox":  [175, 129, 327, 107], "supercategory":  "text",
"category_description":  "SWEET & SPICY PORK RIBS", "bbox":  [54,
239, 568, 145]]
```

*SFT with Composite*

**Input:**
```
Task Description:

Please give me some layout suggestions for drawing a poster
```
**which width and height ratio is 1:1.**  The poster should have the
following text:  **<|text_start|>SWEET & SPICY PORK RIBS<|text_end|>**
**and <|text_start|>Get Saucy!<|text_end|>.**
```
System Prompt:

"NOTE":  "First output your answer about layout generation
within <answer> </answer> tags, then output your reasoning
process and detailed captions including layout information within
<think> </think> tags.  The answer is required by this json
format:  "["supercategory":  "text/logo/underlay/embellishment",
"category_description":  "text slogan(text)/none(logo,
underlay, embellishment)", "bbox":  [x, y, w, h],...]".
The reasoning process needs to infer the bbox data at the
logo/text/underlay/embellishment, and its format follows:
<|bbox_start|>(x,y,w,h)<|bbox_end|><<
logo/text/underlay/embellishment>>.  For text, the text content
needs to be processed as <|text_start|>content<|text_end|><|bbox_
```

start|>(x,y,w,h)<|bbox_end|><<text>>."
**Output:**

**<answer>**["supercategory": "text", "category_description": "Get Saucy!", "bbox": [175, 129, 327, 107], "supercategory": "text", "category_description": "SWEET & SPICY PORK RIBS", "bbox": [54, 239, 568, 145]]**</answer><think>**The poster has the following layout elements: <|text_start|>Get Saucy!<|text_end|><|bbox_start|>(175, 129, 327, 107)<|bbox_end|><<text>> and <|text_start|>SWEET & SPICY PORK RIBS<|text_end|><|bbox_start|>(54, 239, 568, 145)<|bbox_end|><<text>>.**</think>**

**Task for Textual Product Layout**

*SFT*

**Input:**
Task Description:

Please give me some layout suggestions for drawing a poster **which width and height ratio is 1:1** about **<|product_start|>glass mugs<|product_end|>, <|product_start|>cubed sweet potatoes<|product_end|>, and <|product_start|>pork ribs<|product_end|>**, so as to better generate a poster. The poster should have the following text: **<|text_start|>SWEET & SPICY PORK RIBS<|text_end|> and <|text_start|>Get Saucy!<|text_end|>**.
System Prompt:

"NOTE": "You only need output your answer about layout generation. The answer is required by this json format: "["supercategory": "text/product/logo/underlay/embellishment", "category_description": "product name(product)/text slogan(text)/none(logo, underlay, embellishment)", "bbox": [x, y, w, h],...]."
**Output:**

["supercategory": "product", "category_description": "pork ribs", "bbox": [102, 492, 566, 319], "supercategory": "product", "category_description": "glass mugs", "bbox": [2, 370, 367, 313], "supercategory": "product", "category_description": "sweet potatoes", "bbox": [428, 836, 235, 138], "supercategory": "text", "category_description": "Get Saucy!", "bbox": [175, 129, 327, 107], "supercategory": "text", "category_description": "SWEET & SPICY PORK RIBS", "bbox": [54, 239, 568, 145]]

*SFT with Composite*

**Input:**
Task Description:

Please give me some layout suggestions for drawing a poster **which width and height ratio is 1:1** about **<|product_start|>glass mugs<|product_end|>, <|product_start|>cubed sweet potatoes<|product_end|>, and <|product_start|>pork ribs<|product_end|>**, so as to better generate a poster. The poster should have the following text: **<|text_start|>SWEET & SPICY PORK RIBS<|text_end|> and <|text_start|>Get Saucy!<|text_end|>**.
System Prompt:

```
"NOTE":  "First output your answer about layout generation within
<answer> </answer> tags, then output your reasoning process and
detailed captions including layout information within <think>
</think> tags.  The answer is required by this json format:
"["supercategory":  "text/product/logo/underlay/embellishment",
"category_description":  "product name(product)/text
slogan(text)/none(logo, underlay, embellishment)", "bbox":  [x, y,
w, h],...]".  The reasoning process needs to infer the bbox data
at the logo/text/product/underlay/embellishment, and its format
follows:  <|bbox_start|>(x,y,w,h)<|bbox_end|><<
logo/text/underlay/embellishment>>.  For text, the text content
needs to be processed as <|text_start|>content<|text_end|><|bbox_
start|>(x,y,w,h)<|bbox_end|><<text>>."
```
**Output:**

```
<answer>["supercategory":  "product", "category_description":
"pork ribs", "bbox":  [102, 492, 566, 319], "supercategory":
"product", "category_description":  "glass mugs",
"bbox":  [2, 370, 367, 313], "supercategory":  "product",
"category_description":  "sweet potatoes", "bbox":  [428, 836,
235, 138], "supercategory":  "text", "category_description":
"Get Saucy!", "bbox":  [175, 129, 327, 107], "supercategory":
"text", "category_description":  "SWEET & SPICY PORK RIBS",
"bbox":  [54, 239, 568, 145]]</answer><think>The image showcases a
beautifully plated dish featuring a large slab of sweet and spicy
pork ribs<|bbox_start|>(102, 492, 566, 319)<|bbox_end|><<product>>,
adorned with sprigs of fresh cilantro.  Appearing richly
glazed, the ribs are arranged on a white plate with garnishing
around its sides.  Positioned in the background are three glass
mugs<|bbox_start|>(2, 370, 367, 313)<|bbox_end|><<product>>
filled with a light brown beverage.  Complementing the
main dish, small bowls are filled with sautéed or cooked
vegetables such as cubed sweet potatoes<|bbox_start|>(428,
836, 235, 138)<|bbox_end|><<product>>, sliced zucchini with
cherry tomatoes, and a thick red sauce.  Bold text at the top
reads <|text_start|>Get Saucy!<|text_end|><|bbox_start|>(175,
129, 327, 107)<|bbox_end|><<text>> <|text_start|>SWEET &
SPICY PORK RIBS<|text_end|><|bbox_start|>(54, 239, 568,
145)<|bbox_end|><<text>>.</think>
```

**Task for Visual Product Layout**

*SFT*

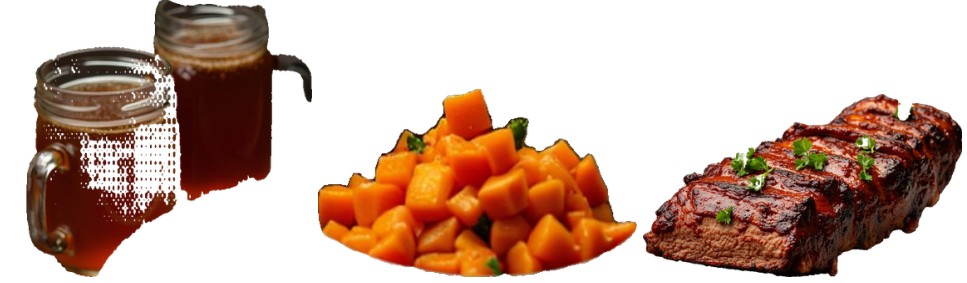

Figure 1: The example input elements of *Task for Visual Product Layout*, denoted by $< image ><$ $image >< image >$ in the task description.

**Input:**
Task Description:

**<image><image><image>Based on the product image I
provided,** please give me some layout suggestions for
drawing a poster **which width and height ratio is 1:1** about
**<|product_start|>glass mugs<|product_end|>, <|product_start|>cubed
sweet potatoes<|product_end|>, and <|product_start|>pork
ribs<|product_end|>,** so as to better generate a poster. The
poster should have the following text: **<|text_start|>SWEET
& SPICY PORK RIBS<|text_end|> and <|text_start|>Get
Saucy!<|text_end|>.**
System Prompt:

"NOTE": "You only need output your answer about layout
generation. The answer is required by this json format:
"["supercategory": "text/product/logo/underlay/embellishment",
"category_description": "product name(product)/text
slogan(text)/none(logo, underlay, embellishment)", "bbox": [x,
y, w, h],...]."
**Output:**

["supercategory": "product", "category_description": "pork
ribs", "bbox": [102, 492, 566, 319], "supercategory": "product",
"category_description": "glass mugs", "bbox": [2, 370, 367,
313], "supercategory": "product", "category_description": "sweet
potatoes", "bbox": [428, 836, 235, 138], "supercategory":
"text", "category_description": "Get Saucy!", "bbox": [175,
129, 327, 107], "supercategory": "text", "category_description":
"SWEET & SPICY PORK RIBS", "bbox": [54, 239, 568, 145]]

*SFT with Composite*

**Input:**
Task Description:

**<image><image><image>Based on the product image I
provided,** please give me some layout suggestions for
drawing a poster **which width and height ratio is 1:1** about
**<|product_start|>glass mugs<|product_end|>, <|product_start|>cubed
sweet potatoes<|product_end|>, and <|product_start|>pork
ribs<|product_end|>,** so as to better generate a poster. The
poster should have the following text: **<|text_start|>SWEET
& SPICY PORK RIBS<|text_end|> and <|text_start|>Get
Saucy!<|text_end|>.**
System Prompt:

"NOTE": "First output your answer about layout generation within
<answer> </answer> tags, then output your reasoning process and
detailed captions including layout information within <think>
</think> tags. The answer is required by this json format:
"["supercategory": "text/product/logo/underlay/embellishment",
"category_description": "product name(product)/text
slogan(text)/none(logo, underlay, embellishment)", "bbox": [x, y,
w, h],...]". The reasoning process needs to infer the bbox data
at the logo/text/product/underlay/embellishment, and its format
follows: <|bbox_start|>(x,y,w,h)<|bbox_end|><<
logo/text/underlay/embellishment>>. For text, the text content
needs to be processed as <|text_start|>content<|text_end|><|bbox_

```
start|>(x,y,w,h)<|bbox_end|><<text>>."
```
**Output:**

**<answer>**["supercategory": "product", "category_description": "pork ribs", "bbox": [102, 492, 566, 319], "supercategory": "product", "category_description": "glass mugs", "bbox": [2, 370, 367, 313], "supercategory": "product", "category_description": "sweet potatoes", "bbox": [428, 836, 235, 138], "supercategory": "text", "category_description": "Get Saucy!", "bbox": [175, 129, 327, 107], "supercategory": "text", "category_description": "SWEET & SPICY PORK RIBS", "bbox": [54, 239, 568, 145]]**</answer><think>**The image showcases a beautifully plated dish featuring a large slab of sweet and spicy pork ribs<|bbox_start|>(102, 492, 566, 319)<|bbox_end|><<product>>, adorned with sprigs of fresh cilantro. Appearing richly glazed, the ribs are arranged on a white plate with garnishing around its sides. Positioned in the background are three glass mugs<|bbox_start|>(2, 370, 367, 313)<|bbox_end|><<product>> filled with a light brown beverage. Complementing the main dish, small bowls are filled with sautéed or cooked vegetables such as cubed sweet potatoes<|bbox_start|>(428, 836, 235, 138)<|bbox_end|><<product>>, sliced zucchini with cherry tomatoes, and a thick red sauce. Bold text at the top reads <|text_start|>Get Saucy!<|text_end|><|bbox_start|>(175, 129, 327, 107)<|bbox_end|><<text>> <|text_start|>SWEET & SPICY PORK RIBS<|text_end|><|bbox_start|>(54, 239, 568, 145)<|bbox_end|><<text>>.**</think>**

**Task for Visual Background Layout**

*SFT*

**Input:**
Task Description:

**<image>Based on the background poster image I provided**, please give me some layout suggestions for drawing a poster **which width and height ratio is 1:1.** The poster should have the following text: **<|text_start|>SWEET & SPICY PORK RIBS<|text_end|> and <|text_start|>Get Saucy!<|text_end|>.**
System Prompt:

"NOTE": "You only need output your answer about layout generation. The answer is required by this json format: "["supercategory": "text/logo/underlay/embellishment", "category_description": "text slogan(text)/none(logo, underlay, embellishment)", "bbox": [x, y, w, h],...]."
**Output:**

["supercategory": "text", "category_description": "Get Saucy!", "bbox": [175, 129, 327, 107], "supercategory": "text", "category_description": "SWEET & SPICY PORK RIBS", "bbox": [54, 239, 568, 145]]

*SFT with Composite*

**Input:**
Task Description:

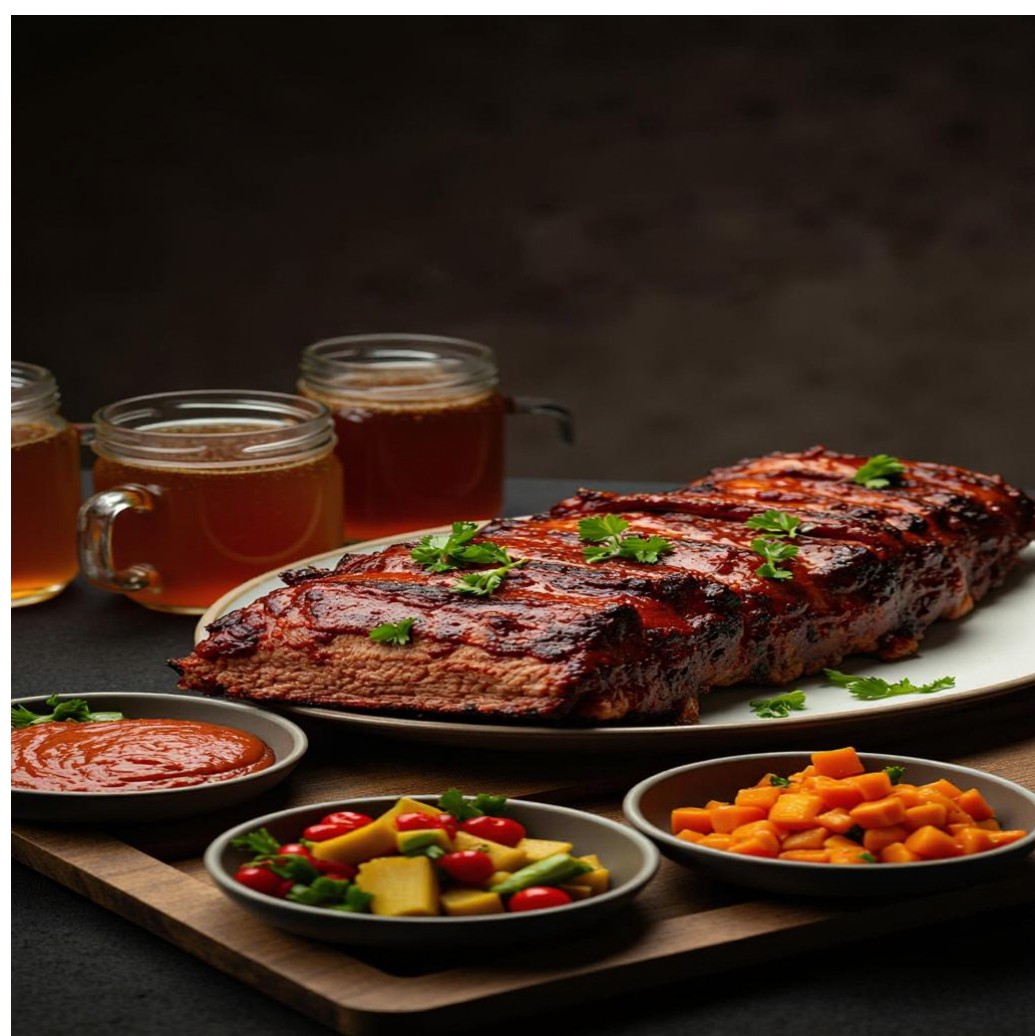

Figure 2: The example input element of *Task for Visual Background Layout*, denoted by $<image>$ in the task description.

**<image>Based on the background poster image I provided**, please give me some layout suggestions for drawing a poster **which width and height ratio is 1:1.** The poster should have the following text: **<|text_start|>SWEET & SPICY PORK RIBS<|text_end|> and <|text_start|>Get Saucy!<|text_end|>.**
System Prompt:

"NOTE": "First output your answer about layout generation within <answer> </answer> tags, then output your reasoning process and detailed captions including layout information within <think> </think> tags. The answer is required by this json format: "["supercategory": "text/ logo/underlay/embellishment", "category_description": "text slogan(text)/none(logo, underlay, embellishment)", "bbox": [x, y, w, h],...]". The reasoning process needs to infer the bbox data at the logo/text/underlay/embellishment, and its format follows: <|bbox_start|>(x,y,w,h)<|bbox_end|><< logo/text/underlay/embellishment>>. For text, the text content

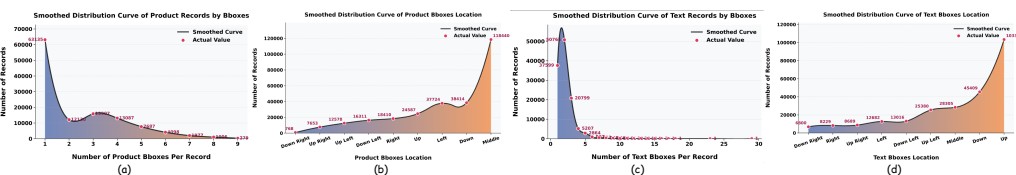

Figure 3: Visualization of the distribution of product and text bboxes in terms of number and location in AnyLayout dataset.

```
needs to be processed as <|text_start|>content<|text_end|><|bbox_
start|>(x,y,w,h)<|bbox_end|><<text>>."
```
**Output:**

```
<answer>["supercategory": "text", "category_description": "Get
Saucy!", "bbox": [175, 129, 327, 107], "supercategory": "text",
"category_description": "SWEET & SPICY PORK RIBS", "bbox": [54,
239, 568, 145]]</answer><think>The image showcases a beautifully
plated dish featuring a large slab of sweet and spicy pork
ribs, adorned with sprigs of fresh cilantro. Appearing richly
glazed, the ribs are arranged on a white plate with garnishing
around its sides. Positioned in the background are three
glass mugs filled with a light brown beverage. Complementing
the main dish, small bowls are filled with sautéed or cooked
vegetables such as cubed sweet potatoes, sliced zucchini with
cherry tomatoes, and a thick red sauce. Bold text at the top
reads <|text_start|>Get Saucy!<|text_end|><|bbox_start|>(175,
129, 327, 107)<|bbox_end|><<text>> <|text_start|>SWEET &
SPICY PORK RIBS<|text_end|><|bbox_start|>(54, 239, 568,
145)<|bbox_end|><<text>>.</think>
```

## A.2 INSTRUCTION ANALYSIS

See the Input Instruction for each four different task examples at Tab. 1a - *Slogan Layout*, Tab. 1b - *Textual Product Layout*, Tab. 2a - *Visual Product Layout* and Tab. 2b - *Visual Background Layout*.

## A.3 DATA ANALYSIS

The proposed dataset, **AnyLayout-120K**, consists of 126,131 annotated records, each containing bounding boxes (bboxes) that classify visual elements into five distinct categories: logo, product, text, underlay and embellishment. The dataset exhibits a high annotation density, with an average of 5.08 bboxes per record, reflecting complex spatial arrangements across the collection. The distribution of bbox counts is concentrated between 4 and 6 per record, representing the majority of instances, while ranging from as few as 2 bboxes (in 4,535 samples) to as many as 32 in highly detailed cases. This long-tailed distribution captures a broad spectrum of image complexity and contextual variation, enhancing the potential for training models with improved generalizability across diverse layout structures described in Fig. 3.

Text localization is a central focus of the dataset, with an average of 2.11 text boxes per record and a total of 151,906 Chinese and 97,744 English text instances. The multilingual composition, characterized by a predominance of Chinese text, renders the dataset particularly suitable for research on multilingual scene text detection and recognition. The mean ratio of text box area to total image area is 0.0407, highlighting the challenge of identifying small-scale textual elements within complex visual layouts. Distribution analysis shows that over 74% of images contain one or two text boxes, while a non-negligible number of samples exhibit higher text densities. This variability supports the investigation of both sparse and dense text configurations, enhancing the dataset's applicability across diverse text detection scenarios.

| # | Input — Slogan Layout |
|---|---|
| 1 | Please propose multiple poster layout options for a canvas with a width:height ratio of <aspect_ratio>. The only required text is: <slogan>. |
| 2 | Suggest composition layouts for a <aspect_ratio> poster featuring the slogan: <slogan>. |
| 3 | Share layout blueprints for a poster (aspect ratio <aspect_ratio>) that highlights the text "<slogan>". |
| 4 | Provide grid-based layout recommendations for a <aspect_ratio> poster. The poster must display: <slogan>. |
| 5 | Outline alternative arrangements for typography and spacing on a <aspect_ratio> poster using the following text: <slogan>. |
| 6 | Give visual hierarchy and placement suggestions for a poster at <aspect_ratio>, centered on the phrase: <slogan>. |
| 7 | List layout concepts (margins, alignment, scale) for a <aspect_ratio> poster whose copy reads: <slogan>. |
| 8 | Recommend headline-first poster layouts for a <aspect_ratio> canvas using the slogan: <slogan>. |
| 9 | Draft several layout sketches in words for a <aspect_ratio> poster; required text content: <slogan>. |
| 10 | Propose minimal-to-bold layout variations for a <aspect_ratio> poster that must include the text: <slogan>. |

(a) Slogan Layout

| # | Input — Textual Product Layout |
|---|---|
| 1 | Please suggest poster layouts (aspect ratio <aspect_ratio>) for advertising <product>. Include the following text: <slogan>. |
| 2 | Provide composition ideas for a <aspect_ratio> sales poster promoting <product> with this copy: <slogan>. |
| 3 | Recommend layout options for a <aspect_ratio> promotional poster for <product>; required text: <slogan>. |
| 4 | Share grid and placement guidelines for a <aspect_ratio> poster aimed at selling <product>, featuring the text: <slog an>. |
| 5 | Outline contrasting layouts for a <aspect_ratio> retail poster for <product>. The poster should read: <slogan>. |
| 6 | Give hierarchy and spacing suggestions for a <aspect_ratio> poster to market <product>, ensuring the text "<slogan>" is prominent. |
| 7 | Propose headline, body, and footer placements for a <aspect_ratio> poster focused on selling <product>; text to show: <slogan>. |
| 8 | List balanced vs. dynamic layout concepts for a <aspect_ratio> poster advertising <product> with the message: <slogan>. |
| 9 | Recommend callout and caption positions for a <aspect_ratio> sales poster for <product>; include: <slogan>. |
| 10 | Draft layout frameworks (rule of thirds, centered, diagonal) for a <aspect_ratio> poster promoting <product> that includes: <slogan>. |

(b) Textual Product Layout

Table 1: Diversified input instructions split into two subtables.

The spatial distribution of textual elements enhances the dataset's diversity, with a predominant concentration along the upper (103,364 instances) and lower (45,409 instances) image boundaries, and substantial representation in central regions (28,305 instances) as well as corner areas. This variability in positioning increases the complexity of text localization, necessitating models to generalize effectively across heterogeneous layouts and spatial configurations shown in Fig. 3.

The dataset exhibits a well-defined representation of primary objects, with an average of 2.30 bboxes per image and a mean box-to-image area ratio of 0.2163, indicating prominent and variably scaled product instances. A clear dichotomy emerges in object distribution: 63,563 images contain a single bounding box, making them suitable for single-object detection tasks, while 56,599 images include multiple objects, supporting research in multi-object detection. Spatial analysis reveals a strong central bias, with 118,440 instances located in the central region; however, substantial object con-

| # | Input — Visual Product Layout |
|---|---|
| 1 | <image> Based on the product image provided, suggest layout options for a <aspect_ratio> poster selling <product>. The poster text should be: <slogan>. |
| 2 | <image> Using this product photo, provide composition and placement recommendations for a <aspect_ratio> promotional poster for <product> that displays: <slogan>. |
| 3 | <image> With the attached product image, outline layout variants for a <aspect_ratio> sales poster; required copy: <slogan>. |
| 4 | <image> Considering the product image, recommend grid systems and focal-point placements for a <aspect_ratio> poster advertising <product>, including the text: <slogan>. |
| 5 | <image> From the product image, propose clean vs. bold layout directions for a <aspect_ratio> poster for <product>. The poster must include: <slogan>. |
| 6 | <image> Analyze the product image and suggest typographic hierarchy and image placement for a <aspect_ratio> poster promoting <product>; use the text: <slogan>. |
| 7 | <image> Please provide hero-shot vs. collage layout ideas for a <aspect_ratio> sales poster for <product> with the message: <slogan>. |
| 8 | <image> Offer safe-zone and margin guidance for arranging the product image in a <aspect_ratio> poster selling <product>. The text content is: <slogan>. |
| 9 | <image> Recommend background, foreground, and overlay placements incorporating the product image in a <aspect_ratio> poster for <product>; include "<slogan>". |
| 10 | <image> Draft wireframe-like layout descriptions that integrate the product image into a <aspect_ratio> poster advertising <product> and showing: <slogan>. |

(a) Visual Product Layout

| # | Input — Visual Background Layout |
|---|---|
| 1 | <image> Using the provided background poster image, suggest layout options for a <aspect_ratio> poster. The required text is: <slogan>. |
| 2 | <image> Based on the background image, provide composition recommendations for a <aspect_ratio> poster that will display: <slogan>. |
| 3 | <image> With this background artwork, outline layout variants for a <aspect_ratio> poster placing the text: <slogan>. |
| 4 | <image> Considering the given background, recommend grid and safe-zone placements for a <aspect_ratio> poster featuring the text: <slogan>. |
| 5 | <image> From the background image, propose minimal vs. layered layout directions for a <aspect_ratio> poster; the copy should read: <slogan>. |
| 6 | <image> Analyze the background and suggest typography, contrast, and placement options for a <aspect_ratio> poster that includes: <slogan>. |
| 7 | <image> Please provide overlay and masking layout ideas for a <aspect_ratio> poster using the background image; required text: <slogan>. |
| 8 | <image> Offer guidance on text legibility and positioning over the background for a <aspect_ratio> poster. The message is: <slogan>. |
| 9 | <image> Recommend focal region, margins, and alignment for placing "<slogan>" on a <aspect_ratio> poster with the supplied background image. |
| 10 | <image> Draft wireframe-style layout descriptions that utilize the background image in a <aspect_ratio> poster and show the text: <slogan>. |

(b) Visual Background Layout

Table 2: Diversified input instructions split into two subtables.

centrations also appear along image boundaries, particularly in the left (37,724) and bottom (38,414) regions. This heterogeneity in object quantity and spatial placement enhances the dataset's utility for developing and evaluating robust object localization and recognition models under diverse and challenging conditions demonstrated on Fig. 3.

## A.4 MORE RESULTS VISUALIZATION

As shown in Fig. 4, 5, 6, and 7, we visualized more diverse generation results under different tasks.

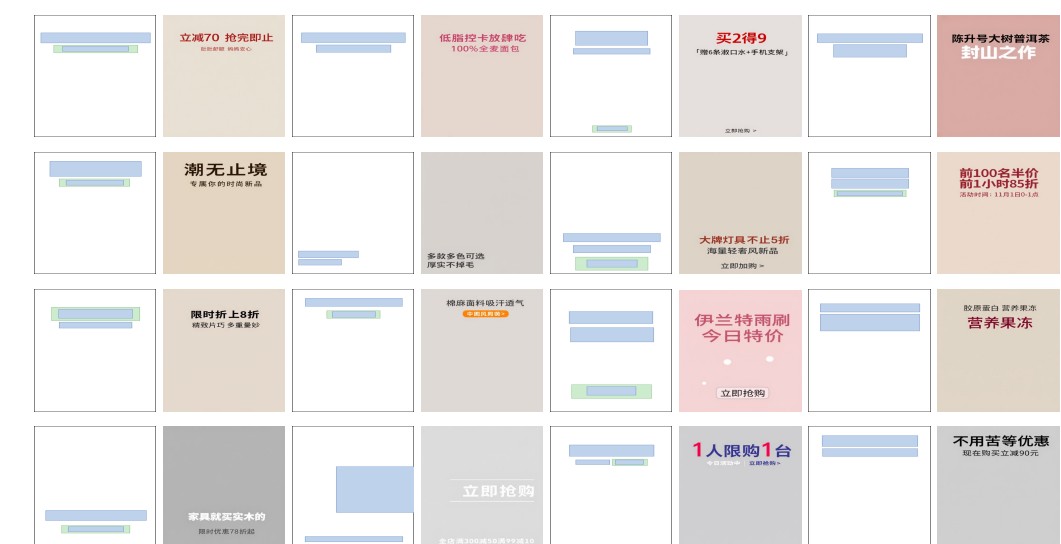

Figure 4: More results on Slogan Layout Task.

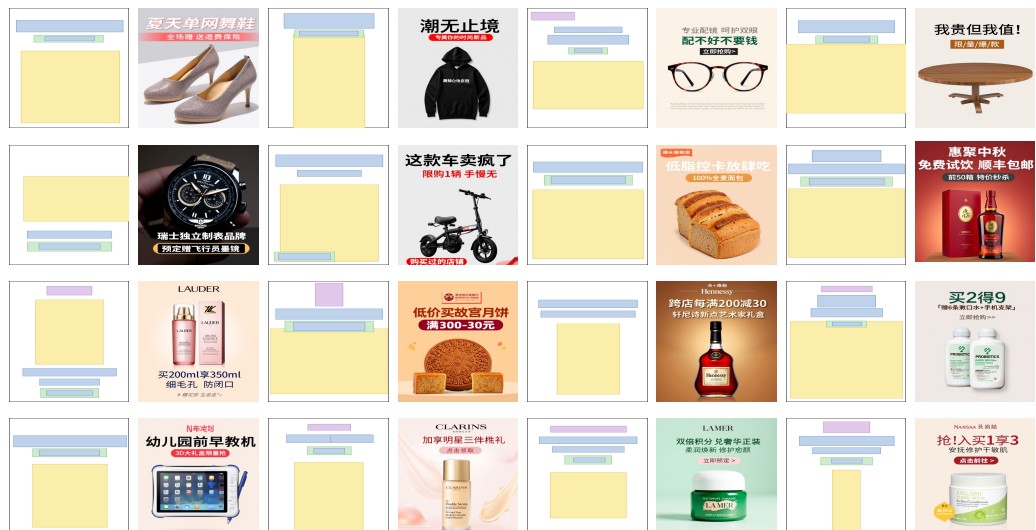

Figure 5: More results on Textual Product Layout Task.

## A.5 LLM CLARIFICATION

We clarify the use of Large Language Models (LLMs) in the preparation of this manuscript. Specially, LLMs were employed for language polishing and AnyLayout-120K composite and instruction generation. This process involved correcting grammatical errors, improving sentence structure, and enhancing the overall readability and flow of the text. And as we described in main paper Sec. 3.3, we clarify how we utilize VLM/LLM to generate composite and instruction which is check by human reviewer and no harm to the society. It is crucial to emphasize that all core scientific content, intellectual contribution, and original ideas presented in this paper are exclusively the work of the human authors. This includes the formulation of the research problem, the development of the AnyLayout framework, the experimental design, and the analysis of the results. The LLM served strictly as a writing aid and was not involved in any conceptual or analytical aspect of this research.

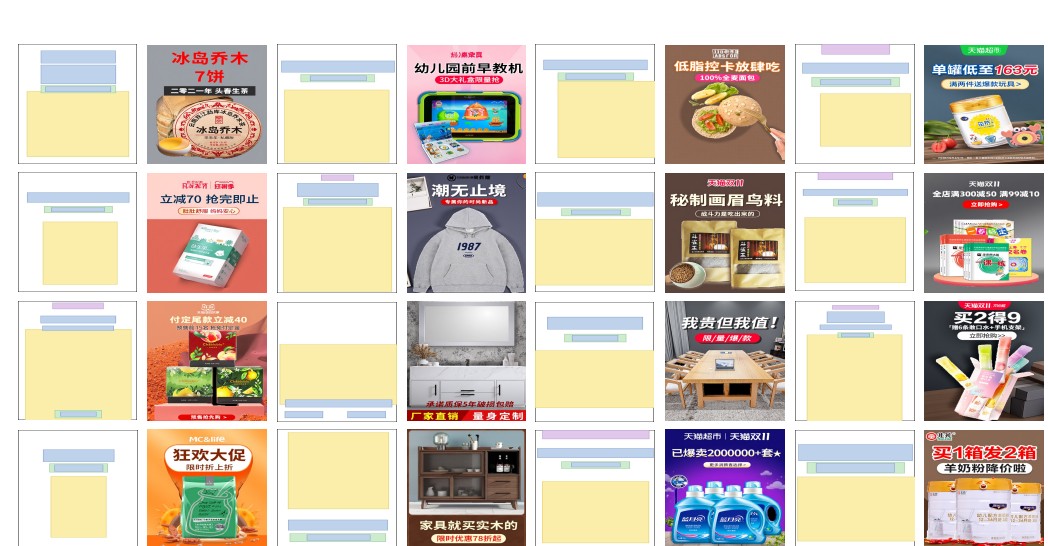

Figure 6: More results on Visual Product Layout Task.

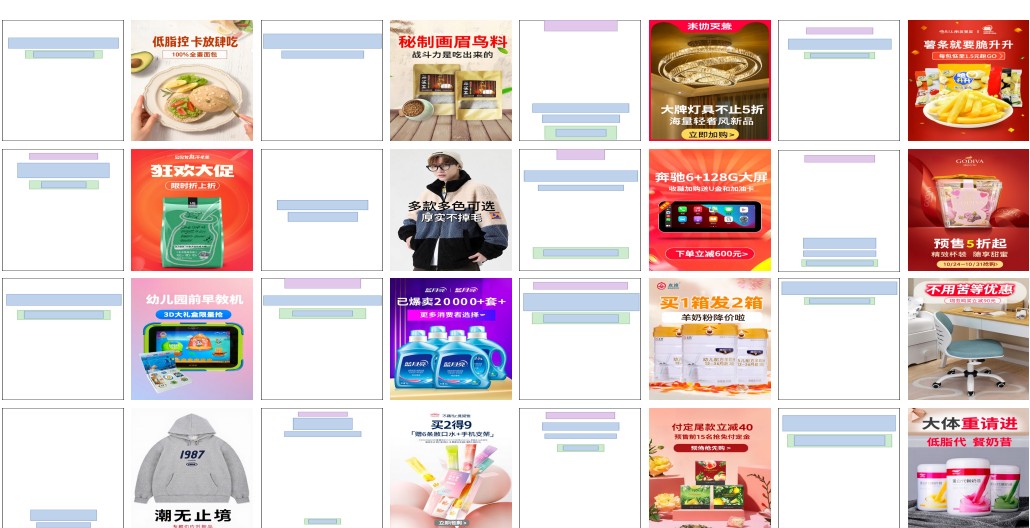

Figure 7: More results on Visual Background Layout Task.