# OpenReview forum: "AnyLayout: Versatile Advertising Poster Layout Generation with MLLMs"
_ICLR.cc/2026/Conference — ICLR 2026 Conference Withdrawn Submission_

### Official Review · Reviewer_F6mx · 2025-10-30

**Soundness:** 2
**Presentation:** 2
**Contribution:** 2
**Rating:** 2
**Confidence:** 4

**Summary:**

This paper focuses on 4 layout generation tasks, including solgan layout generation, textual product layout generation, visual product layout generation, and visual background layout generation. A new dataset, i.e., AnyLayout-120k, is proposed with detailed annotations. Several evaluation metrics are further adopted to ensure the layout quality assessment.

**Strengths:**

The proposed dataset construction process is a reasonable solution for improving the layout generation performance. The released dataset could be useful for future research in the community.

**Weaknesses:**

1. The motivation of this paper is to create new datasets to support multiple layout generation tasks. However, it is somewhat overclaimed. Many existing works can also support different layout generation tasks in a unified model.
2. The technical contribution of this paper is relatively limited. While the proposed dataset belongs to a large-scale one, all data comes from the combination of existing public datasets (CGL, PKU, AutoPoster, CreatiDesign). In addition, the metrics used in this paper belong to commonly used design rules. I cannot see new insights from the dataset construction and evaluation processes.
3. The experimental evaluations are not thorough. The current evaluations only compare with two LLM-based methods, lacking detailed evaluations with closely related works, e.g., [1-5].

[1] Zhang, Hui, et al. "Creatidesign: A unified multi-conditional diffusion transformer for creative graphic design." arXiv preprint arXiv:2505.19114.

[2] Zhang, Hui, et al. "Creatilayout: Siamese multimodal diffusion transformer for creative layout-to-image generation." ICCV 2025.

[3] Horita, Daichi, et al. "Retrieval-augmented layout transformer for content-aware layout generation." CVPR 2024.

[4] Hsu, HsiaoYuan, and Yuxin Peng. "PosterO: Structuring Layout Trees to Enable Language Models in Generalized Content-Aware Layout Generation." CVPR 2025.

[5] Chen, Haoyu, et al. "Posta: A go-to framework for customized artistic poster generation." CVPR 2025.

**Questions:**

1. This paper claims that the proposed formulation is model-agnostic. How about the performance of the proposed method on different backend MLLMs?
2. How about the details of the Similarity, IoU, and Human Check process? Is each case checked by one human reviewer?

---

### Official Review · Reviewer_p7bZ · 2025-11-01

**Soundness:** 2
**Presentation:** 2
**Contribution:** 2
**Rating:** 2
**Confidence:** 4

**Summary:**

In this paper, the authors study how to map user intentions into poster layouts. The condition includes textual descriptions, product images, background images, and so on. To achieve this, they construct AnyLayout-120K, which contains pairwise (instruction, layout) samples and covers four layout generation tasks: slogan layout, textual product layout, visual product layout, visual background layout generation. In addition, they develop 7 geometry-aware evaluation metrics specifically designed for product-centered poster design. Using AnyLayout-120K, the authors establish a strong baseline using Qwen2.5-VL-7B through supervised fine-tuning, achieving state-of-the-art layout generation performance.

**Strengths:**

The paper clearly articulates limitations of existing approaches and motivates the need for flexible, instruction-driven layout generation. By introducing novel product-centered metrics, the work provides a comprehensive evaluation on layout quality. Qualitative results across the 4 tasks demonstrate the effectiveness of AnyLayout.

**Weaknesses:**

- Qualitative comparison is not included. The authors only visualize the qualitative results of AnyLayout. However, it is also necessary to show the results of baseline methods, such as PosterLlama and PosterLLaVA.
- No discussion or comparison with recent powerful image generators. For example, GPT-4o and Gemini can also produce visually appealing posters from user instructions. Does AnyLayout perform better than these models?
- Is it a fair comparison to evaluate PosterLlama and PosterLLaVA in zero-shot mode? To be more specific, the proposed model is trained on 4 datasets (PKU, AP, CGL, CD), while the two baseline models only see some of them. Although the authors explain the I/O format mismatch issue on line 376, it is still not convincing enough. Why not unify the I/O formats across different datasets and retrain these baselines?
- Regarding the dataset construction, Figure 2 shows that some visual elements are not fully shaped (e.g., fruits). How do the authors address these issues, or do the authors not perform additional processing? While it is understandable (due to occlusion between elements), including such data may severely degrade the dataset quality.
- The technical contributions of the work are relatively limited. The model is standard SFT on Qwen2.5-VL without novel designs. The main contribution is the dataset and task formulation.
- Although the work introduces novel metrics, they are mainly geometric and rule-based. Aesthetic quality assessment of the output posters is missing (e.g., user study, VLM judge).

**Questions:**

Please see the weaknesses.

---

### Official Review · Reviewer_bAPd · 2025-11-03

**Soundness:** 2
**Presentation:** 2
**Contribution:** 2
**Rating:** 2
**Confidence:** 4

**Summary:**

This paper introduces AnyLayout-120K, a large-scale instruction-driven dataset for advertising poster layout generation, and proposes an MLLM-based layout generation model built upon Qwen2.5-VL. The authors define four sub-tasks (Slogan Layout, Textual Product Layout, Visual Product Layout, and Visual Background Layout)  and claim that their formulation extends layout generation from fixed-canvas bounding-box prediction to versatile, instruction-driven settings. The paper also presents seven geometry-aware evaluation metrics for assessing layout quality.

**Strengths:**

**1. Comprehensive dataset curation.**

The authors combine several existing datasets (PKU, CGL, AutoPoster, CreatiDesign) into a unified benchmark with 120K samples, offering a consistent structure for evaluating multimodal layout generation tasks.

**2. Clear experimental organization.**

The paper provides extensive quantitative results and ablation studies across multiple baselines, including PosterLlama and PosterLLaVA, along with newly proposed metrics. The experimental section is well structured.

**3. Consistency in data–metric–model pipeline.**

The data construction process, the geometry-aware metrics, and the MLLM-based model share a coherent design philosophy focused on spatial precision and controllability.

**Weaknesses:**

**1. Limited novelty in task formulation.**

Despite the claim of introducing a “versatile, instruction-driven” layout generation task, the actual formulation only revisits bounding-box prediction for existing subtasks (e.g., slogan placement, product alignment). The work does not go beyond layout-level reasoning or full poster synthesis.
→ Prior works such as PosterLayout, AutoPoster, PosterLlama, and PosterLLaVA already addressed similar subtasks under more comprehensive frameworks.

**2. Dataset largely repurposes existing resources.**

AnyLayout-120K is mainly a recombination of existing datasets with minor instruction rewriting. It does not introduce new design domains, visual styles, or genuinely user-driven layout intentions. The dataset’s incremental contribution appears insufficient for a new benchmark paper.

**3. Trivial task difficulty.**

The paper only evaluates layout-level box generation, which is far simpler than current end-to-end layout-to-image or design synthesis tasks. The task complexity is low, and the proposed evaluation metrics focus solely on geometric alignment rather than perceptual or aesthetic quality.

**4. Weak modeling contribution.**

The proposed model is a fine-tuned version of Qwen2.5-VL. There are no architectural innovations  (the method essentially serializes JSON-like layout data for supervised fine-tuning). The claimed improvements are likely due to dataset size and instruction tuning rather than algorithmic novelty.

**5. Outdated research direction.**

With the rapid progress of end-to-end poster generation frameworks, such as Planning and Rendering (Li et al., 2023), PosterMaker (Gao et al., CVPR 2025), and POSTA (Chen et al., CVPR 2025), modern systems can already synthesize complete poster images directly from multimodal inputs, integrating layout planning, background rendering, and text composition in a unified pipeline.
In contrast, this paper focuses solely on layout-level box prediction, without engaging with the visual realization or style aspects that dominate current poster generation research.
As a result, the proposed dataset and benchmark feel out of step with current research trends, offering limited practical value or novelty in the context of today’s diffusion- and LLM-based end-to-end design automation landscape.

**6. Evaluation fairness issues.**

Competing baselines (PosterLlama, PosterLLaVA) are evaluated in zero-shot mode, while the proposed model is fine-tuned on the target dataset, which makes the reported gains difficult to interpret as genuine superiority.

**Questions:**

1. How do you justify the need for a layout-only benchmark when most recent poster generation works handle end-to-end image synthesis including layout, texture, and style?

2. Could you elaborate on how your “instruction-driven” component goes beyond simply adding textual prompts? Are there examples of compositional or multi-turn reasoning?

3. Have you evaluated whether the proposed geometry-aware metrics correlate with human aesthetic judgments?

---

### Official Review · Reviewer_crb6 · 2025-11-04

**Soundness:** 3
**Presentation:** 3
**Contribution:** 3
**Rating:** 4
**Confidence:** 3

**Summary:**

This paper proposes AnyLayout-120K, a large-scale instruction-driven dataset for multimodal advertising poster layout generation, along with 7 geometry-aware evaluation metrics and a unified baseline model based on multi-modal large language models (MLLMs).

**Strengths:**

1. This paper proposes the AnyLayout-120K Dataset, a high-quality, annotated, large-scale dataset that covers multiple layout generation tasks and is beneficial to the future development of the layout generation field.
2. This paper introduces 7 geometry-aware metrics to assess spatial accuracy and compliance with design principles from dimensions such as product centrality, size ratio, and occlusion. They are more comprehensive than traditional metrics.
3. This paper establishes a baseline model that generates layouts in natural language-formatted composite sequences. It supports four sub-tasks, leverages cross-task synergies via multi-task training, and achieves state-of-the-art performance.

**Weaknesses:**

The workload of this article is evident, and the dataset and metrics make significant contributions to the future development of the field. However, the method for constructing the baseline seems overly simple and intuitive, failing to propose obvious improvements related to the characteristics of the poster layout generation task.

**Questions:**

1. It is recommended to supplement subjective evaluation results of layouts generated by this method and comparative methods for more comprehensive validation.
2. It is suggested to further elaborate on both the methodological innovations and limitations of the paper.

---

### Note · Authors · 2025-12-03

I have read and agree with the venue's withdrawal policy on behalf of myself and my co-authors.